# Gene Editing by Extracellular Vesicles

**DOI:** 10.3390/ijms21197362

**Published:** 2020-10-05

**Authors:** Dmitry Kostyushev, Anastasiya Kostyusheva, Sergey Brezgin, Valery Smirnov, Elena Volchkova, Alexander Lukashev, Vladimir Chulanov

**Affiliations:** 1National Medical Research Center of Tuberculosis and Infectious Diseases, Ministry of Health, 127994 Moscow, Russia; kostyusheva_ap@mail.ru (A.K.); Seegez@mail.ru (S.B.); vladimir@chulanov.ru (V.C.); 2Institute of Immunology, Federal Medical Biological Agency, 115522 Moscow, Russia; vall@mail.mipt.ru; 3Sechenov First Moscow State Medical University, 119146 Moscow, Russia; az@rcvh.ru (E.V.); alexander_lukashev@hotmail.com (A.L.)

**Keywords:** gene editing, biodistribution, pharmacokinetics, nanomedicines, nanovesicles, exosomes, nanoparticles, nanoblades, stem cells, mesenchymal stem cells

## Abstract

CRISPR/Cas technologies have advanced dramatically in recent years. Many different systems with new properties have been characterized and a plethora of hybrid CRISPR/Cas systems able to modify the epigenome, regulate transcription, and correct mutations in DNA and RNA have been devised. However, practical application of CRISPR/Cas systems is severely limited by the lack of effective delivery tools. In this review, recent advances in developing vehicles for the delivery of CRISPR/Cas in the form of ribonucleoprotein complexes are outlined. Most importantly, we emphasize the use of extracellular vesicles (EVs) for CRISPR/Cas delivery and describe their unique properties: biocompatibility, safety, capacity for rational design, and ability to cross biological barriers. Available molecular tools that enable loading of desired protein and/or RNA cargo into the vesicles in a controllable manner and shape the surface of EVs for targeted delivery into specific tissues (e.g., using targeting ligands, peptides, or nanobodies) are discussed. Opportunities for both endogenous (intracellular production of CRISPR/Cas) and exogenous (post-production) loading of EVs are presented.

## 1. Introduction

Recent progress in gene-editing technologies, including zinc-finger nucleases (ZFNs), transcription activator-like effector nucleases (TALENs), engineered meganucleases, and clustered regularly interspaced short palindromic repeat (CRISPR)/CRISPR-associated protein (Cas) nucleases, has greatly expanded the opportunities for accurate disruption or modification of the target genomic locus [1]. Here, we review the opportunities for extracellular vesicle (EV)-based delivery of CRISPR/Cas as a robust tool with the most rapidly expanding use.

CRISPR/Cas-based gene editing is a prominent, recently developed molecular technique that has already revolutionized biology and could dramatically transform clinical management of genetically defined conditions, including cancer, infectious, and genetic diseases [2]. CRISPR/Cas systems function by recruiting the Cas protein to a specific locus on a DNA or RNA molecule using a short RNA called single guide RNA (sgRNA) [3]. The Cas protein then introduces a break into the targeted nucleic acid [4,5]. Alternatively, nuclease-null Cas9 (or dead Cas9; dCas) proteins may serve as carriers to bring enzymes or functionally active factors to certain locations in the genome [6,7]. CRISPR/Cas systems provide the powerful means to directly modify genetic, epigenetic, and protein-based pathogenic mechanisms, projecting their application for treating numerous diseases [8,9].

So far, one of the major challenges is the lack of an optimized tissue-specific CRISPR/Cas delivery tool [10]. Many nanotechnological vehicles have been devised in recent years to deliver CRISPR/Cas systems into target cells (reviewed in [11,12,13,14]). Current strategies have numerous limitations, including: (1) high molecular mass and positive charge of Cas proteins that make them difficult to package using common drug delivery tools [15]; (2) the lack of robust tissue-specific delivery vehicles suitable for cell-specific gene editing applications [16]; (3) immunogenicity [17,18,19,20,21] and other safety issues (molecular, cellular and tissue toxicity) [22] to which the majority of novel synthetic delivery vehicles are prone; and, finally, (4) the lack of a universal CRISPR/Cas delivery platform that can be utilized for a wide array of CRISPR/Cas systems. Such a platform must allow use of CRISPR/Cas systems that are highly variable in size and molecular features [23]; systems isolated from various species (e.g., *Neisseria meningitides* [24], *Streptococcus thermophiles* [25], *Streptococcus pyogenes* [4], and others such as the recently described small CasX from Deltaproteobacteria [26]); and engineered CRISPR/Cas [27], such as CRISPRa/i tools [28,29,30], CRISPR base editors [31,32], and the PrimeEditing system [33]. The lack of robust and safe CRISPR/Cas delivery tools, especially with tissue-targeting modalities, delays translation of CRISPR/Cas-based therapeutics into the clinic. In particular, CRISPR/Cas systems have been shown to be highly potent antivirals eliminating or dramatically reducing viral loads in such infections as hepatitis B virus [34,35,36], hepatitis C virus [37], human immunodeficiency virus (HIV) [38,39,40], human papillomavirus [41], and even the recently emerged coronaviral SARS-CoV-2 infection [42]. Notably, CRISPR/Cas systems have been successfully leveraged to genetically modify the human genome for making primary CD4^+^ T cells resistant to HIV [43]; several ongoing clinical trials are underway using CRISPR/Cas for correcting mutations associated with genetic disorders and treating cancer.

Three principal methods are available to deliver Cas and their guiding RNAs (gRNAs) into target cells: (1) coding DNA sequences; (2) coding RNA/mRNA; and (3) ribonucleoprotein complexes (RNPs), i.e., readily available Cas protein complexes with in vitro-transcribed or synthetically generated gRNAs. Delivery of coding DNA sequences can be performed by both viral (including adeno-associated virus and adenovirus) and non-viral methods; packaging and delivery of mRNA/RNA and RNPs are usually performed by non-viral methods [16]. Nanotechnological methods mostly rely on the use of liposomes and cationic lipids [44,45,46], amphiphilic peptides [47], DNA nanoclews [48,49], gold nanoparticles [50,51,52], and graphene-based nanosheets [53].

Delivering CRISPR/Cas as DNA coding sequences is fraught with poorly controllable intracellular synthesis of CRISPR/Cas components with an ensuing increase in off-target activity [54,55,56] and potential integration of DNA into the genome [57]. Although plenty of novel approaches have been proposed to hone the specificity of CRISPR/Cas systems (e.g., self-inactivating delivery systems [58,59], on/off-inducible systems [60,61]) and build additional levels of tunability (e.g., anti-CRISPR proteins [62,63]), these approaches add complexity and safety issues. Delivering large amounts of DNA is also associated with toxicity, may induce activation of the host factors involved in foreign DNA recognition, and may even cause cell death [64,65,66,67]. Additionally, the large molecular size of traditional CRISPR/Cas nucleases and, especially, dCas-based molecular tools exceeds the packaging capacity of commonly used AAV viral vectors and thus hampers their use. This is particularly true for hybrid CRISPR/Cas systems fused to additional functional moieties (epigenome modifiers, transposases [68,69], reverse transcriptases [33], etc.), that add molecular weight to Cas proteins.

Delivery of CRISPR/Cas as mRNA/RNAs is associated with instability and fragility of the long Cas mRNAs and may be substantially compromised by reduced efficacy of on-target editing [70,71,72]. The most straightforward approach is direct delivery of CRISPR/Cas RNPs into the cells [73]. Successful gene editing for treating a disease, whether a genetic disorder or an infectious illness, usually requires very transient expression of CRISPR/Cas, which may permanently correct the malfunctioning gene or rapidly destroy the viral genomes. Many recent studies demonstrated that the delivery of CRISPR/Cas RNPs is characterized by the highest efficacy and specificity of gene editing [74,75,76].

Proteins or RNPs cannot be delivered systemically as naked molecules. Human serum contains proteases that can rapidly destroy unprotected proteins. Protein and RNA components of CRISPR/Cas are therefore vulnerable to rapid degradation upon systemic injection and must be protected by nanoparticles for in vivo applications.

Moreover, pre-existing antibodies against Cas proteins [17] and immune response to Cas and sgRNAs [18] can limit efficacy of CRISPR/Cas approaches. Reducing and evading immune recognition can be achieved by rationally designing Cas proteins (e.g., epitope masking or limiting presentation of Cas epitopes to the immune system) [77], using CRISPR/Cas systems from non-pathogenic organisms, inducing immune tolerance [78], or shielding Cas proteins in systemic circulation. Short-lived CRISPR/Cas complexes are sufficient for most clinical applications, especially in immune-privileged organs, and are less likely to induce a meaningful immune response. Nevertheless, in order to increase efficacy and preserve a second-use opportunity, it is desirable to shield Cas and RNPs from immune recognition.

Cas proteins are not naturally able to cross biological barriers without specially designed delivery vehicles and do not tend to accumulate in specific organs or tissues. High positive charge and molecular mass (>160 kDa for *S. pyogenes* Cas9) make CRISPR/Cas RNPs unsuitable for traditional methods of nanotechnological packaging and protein delivery. Thus, engineering advanced molecular vehicles encapsulating the CRISPR/Cas RNPs with penetrating and targeting ability is needed for tissue-specific delivery of gene editing complexes and clinical implementation of devised molecular techniques.

The ideal non-viral method for targeted in vivo drug delivery should fit the following criteria: (a) effectively package CRISPR/Cas RNPs of any type and species and with any modifications; (b) shield RNPs from an aggressive environment and the immune system; (c) effectively deliver RNPs into target organs; (d) escape endolysosomal pathways; and (e) be simple and scalable. To date, no such methods exist.

To date, the most successful approach for local targeted delivery of gene editing systems in the form of gold-linked nanoparticles combined with penetrating peptides demonstrated ~30% efficacy [50]. However, this method does not shield Cas proteins and suffers from other disadvantages, such as immunogenicity, toxicity, and rapid clearance upon systemic administration. On the other hand, EVs have emerged as a promising delivery system for proteins and RNAs, substantially outperforming synthetic nanocarriers in terms of safety and pharmacokinetics [79]. Evs are natural nanoparticles secreted by numerous cell types that exhibit very high biocompatibility and extraordinary ability to cross biological barriers [80]. Because Evs can transfer RNA, protein, and lipid cargo, display preferential tropism for certain tissues, and are amenable to engineering, they have been extensively utilized as potential drug delivery systems. Genetic engineering of EV-producing cells and modification of purified Evs enables direct loading of therapeutic macromolecules into the vesicles and targeted drug delivery.

The advantages of Evs have been increasingly utilized for CRISPR/Cas delivery, but translating EV-CRISPR/Cas therapies to the clinic requires the invention of new, more efficient techniques for EV cargo loading and surface engineering. Overall, there is great demand for developing effective, programmable, versatile, and safe delivery platforms that ideally can be used for any type of CRISPR/Cas system.

In this manuscript, we discuss the benefits of using EVs for gene-editing applications, and review available technologies for engineering EVs and the recent progress in CRISPR/Cas RNPs delivery using synthetic nanoparticles and EVs, synthetic or naturally produced.

## 2. Extracellular Vesicles (EV) as Drug Delivery Vehicles

### 2.1. General Characteristics

EVs are phospholipid bilayer nano-sized vesicles that are shed by many cell types in the human body. Unlike synthetic nanocarriers, EVs have a membrane that strongly resembles the natural cytoplasmic cell membrane and contains transmembrane and membrane-anchored proteins that facilitate their uptake by target cells and promote the release of the payload into the target tissue. These properties, along with others reviewed in this manuscript (see Section 2.2, Section 2.3 and Section 2.4), make EVs invaluable for developing novel nanotherapeutics and, prospectively, delivery vehicles for CRISPR/Cas tools.

EVs vary in size, structure, vesicle composition, and packaged cargo [81]. Based on the mechanism of biogenesis, EVs are categorized into three major populations: exosomes, microvesicles, and apoptotic bodies [82]. Exosomes are released as the result of fusion of multivesicular bodies, generated by inward budding of the endosomal limiting membrane, with the plasma membrane. The size of exosomes ranges from 30 to 100 nm. Exosomes carry many different types of RNA (micro RNA, vault RNA, tRNA, mRNA, etc.), proteins (tetraspanins, integrins, immunoglobulins, cytoskeletal proteins, heat shock proteins, etc.), and lipids (phosphatidylserine, lysophosphatidylcholine, eicosanoids, etc.), and are important in the interplay between cell populations in living organisms [83]. Microvesicles are formed by direct outward budding of the plasma membrane and are generally larger and more heterogeneous in size and particle composition than exosomes. Sorting of protein and RNA cargo into exosomes is a delicately regulated process, involving sophisticated cellular machinery; in contrast, microvesicles represent an enclosed fraction of cell periphery. Finally, apoptotic bodies are large vesicles, 500–2000 nm in size, produced due to the disassembly of apoptotic cells, and contain randomly distributed remnants of the dying cells. As the nomenclature of EVs is rapidly evolving and the specific type of particles used in a certain study varies based on isolation and characterization methods, for consistency we use the general term “EVs” to describe cell-derived nanoparticles.

### 2.2. Extracellular Vesicles (EV) as Drug Delivery Vehicles

EVs carry many functional biomolecules and are particularly enriched in certain protein pools and classes of nucleic acids. They can naturally carry functional biomolecules implicated in intercellular communication and involved in disease pathogenesis and are able to serve as therapeutic entities delivering functional RNAs, proteins, or lipids [84,85]. EVs are highly biocompatible, as they originate from a natural source. They shield the cargo in a phospholipid bilayer membrane, protecting it from the exogenous environment. Intravascular enzymes (nucleases and proteases) do not affect EV cargo composition and do not impair EV trafficking into target sites [86].

EVs have an intrinsic ability to efficiently cross natural biological barriers and deliver their cargo into target cells when administered systemically. In contrast to synthetic nanoparticles, EVs effectively enter circulation upon injection but are resistant to the major factors contributing to a short half-life of synthetic drugs, such as clearance by renal filtration, opsonization by plasma proteins, or sequestration by mononuclear phagocyte system [87,88].

Surface composition, size, and charge determine EV kinetics in circulation and biodistribution [89]. One of the most prominent biological barriers for systemic nanotherapeutics is the endothelial barrier, i.e., the endothelial cells that line the vasculature [90]. A very compact organization of endothelial cells physically hinders nanoparticles from penetrating into tissues. Gaps in the endothelial wall are several hundred nanometers (0.1–2 µm) for different tissues and conditions, but this cutoff can be dramatically altered in pathologic conditions. Cancer and inflammation alleviate many biological barriers (reduce extracellular matrix, increase vascularization of tissues, promote receptor-mediated recognition of nanovesicles), and impair endothelial barriers by substantially increasing the size of endothelial gaps [91,92,93]. This has implications for therapeutic interventions, allowing the passage of nanotherapeutics through the endothelial wall in affected tissues. Enhanced permeability and retention of nanoparticles can account for >20% increase in nanoparticle accumulation compared to healthy organs [94]. On the other hand, complications of human diseases, such as fibrosis and aberrant circulation, can markedly reduce blood flow and the access of nanoparticles to organs [95].

The most restrictive endothelial organization is the blood-brain barrier, which hinders ~98% of available therapeutics from entering the brain [96]. Crossing the blood-brain barrier is a major challenge for delivering nanotherapeutics. However, the high biocompatibility of EVs enables them to successfully cross this barrier [97].

Extravasation across the endothelial barrier is not the last obstacle before nanotherapeutics reach their final destination. In physiological and certain disease conditions, cells generate a dense extracellular matrix that impedes large nanoparticles from reaching target cells [98]. Finally, nanoparticles have to escape or avoid the endolysosomal compartment [99] and release functional cargo into the cytosol or cell nuclei. The low pH and abundance of hydrolases in endolysosomes affect CRISPR/Cas RNPs, destabilizing the complex and degrading the individual components. Escaping the endolysosomal pathway is critically important for successful CRISPR/Cas delivery into the target cells. Endolysosomal escape can be enhanced by introducing specific ligands onto the surface of EVs, such as pH-sensitive GALA peptides [100] or melittin [101]; using a combination of delivery vehicles with compounds such as chloroquine, imidazole, and zinc chloride [102]; introduction of pH-sensitive constituents (PLGA, NH_4_HCO_3_) [103]; and other strategies.

Upon cytoplasmic delivery, trafficking CRISPR/Cas RNPs into the nucleus is an easy task, as Cas proteins are provisionally linked with nuclear localization signals that effectively promote nuclear transfer of CRISPR/Cas RNPs. When transported into the nucleus, CRISPR/Cas RNPs may evoke an epigenomic or genomic perturbation in target cells [27].

Direct comparison of synthetic nanoparticles and EVs demonstrated that synthetic nanocarriers are internalized by cells less efficiently. The natural composition of EVs may enable their efficient fusion with the plasma membrane or internalization by phagocytosis, micropinocytosis, or endocytosis [104,105]. Uptake of EVs by target cells also relies on various biological ligands and receptors on the surface of EVs. In contrast, only a minor fraction of synthetic nanoparticles enter cells, with the majority accumulating into large aggregates at the cell surface [106].

A plethora of studies demonstrate efficient delivery of naturally occurring RNA and protein cargo by EVs upon systemic administration. EVs derived from CCR5-positive monocytes increase CCR5 expression in target cells, facilitating infection with HIV [107]. Delivery of EVs isolated from an aggressive cancer cell line induced a highly metastatic phenotype in target cells [108]. Similarly, loading EVs successfully transferred cargo into the brain [97]. EVs produced by mesenchymal stem cells (MSCs) promoted cardiac regeneration after myocardial infarction and induced regeneration of tissues in other inflammatory and trauma conditions [109]. EV-mediated delivery was implemented for knocking down PTEN in breast cancer cells with miRNA19 and reducing KRAS levels with siRNA in a pancreatic cancer cell model [110,111]. Overall, EVs can efficiently cross biological barriers, enter target cells, and release functional cargo.

### 2.3. Safety of EVs

Synthetic nanoparticles are typically generated using compounds that may be toxic or immunogenic when injected into the body [112]. In contrast, EVs are generated from a natural source, and if the production source is the same as the recipient species (e.g., using human cells to produce exosomes to be administered to humans), then the obvious potential hazards for the recipients may be related to the major histocompatibility complex (MHC) molecules in EVs presented to the immune system. Upon allogeneic transplantation of EVs, in parallel to allogeneic transplantation of stem cells, the issue of MHC histocompatibility may potentially induce immunogenic response and rejection of EVs (with ensuing decline in therapeutic efficacy) or immune response against tissues that uptook EVs [113,114,115]. Expression of foreign MHCs can induce proliferation of specific T cells and stimulation of the immune response against the MHC-displaying particles [116]. Therefore, injection of non-MHC-matched EVs may potentially induce an immune response in the recipient, damaging or even destroying the cells that engulf EVs. This is particularly true for therapies involving mesenchymal stem/stromal cells (MSCs), which originally were thought to express no or minimal amount of MHC antigens on their surface [117]. A compelling line of evidence has indicated that injecting MSCs into allogeneic donors induces immune response, resulting in rapid clearance of MSCs and low efficacy of therapeutic interventions. In contrast, several studies demonstrated that EVs are hypoimmunogenic or much less immunogenic than EV-producing cells per se [111,118,119]. The current data suggest that although MSCs can exhibit immunosuppressive and immunomodulatory properties, they are not intrinsically immunoprivileged and can trigger immune responses [113,117].

Injecting MSC-derived EVs in vivo, for example, did not elicit an immune rejection of target cells and even suppressed immunity [120]. Moreover, MSC-derived EVs inhibited antigen presentation by immunocompetent cells, reduced expression of MHC molecules and costimulatory molecules, and suppressed inflammation (via attenuated production of inflammatory cytokines, reduced activation of T cells, etc.) [121,122,123,124]. These studies imply the possibility of using EVs as customizable nanoparticles suitable for allogeneic implantation. Although EVs also pack MHC antigens inside the vesicles, their abundance is significantly lower compared to that of the cell of origin [125]. Producing EVs in cell types that avoid or partly avoid immune rejection, such as MSCs, considerably reduces their immunogenicity and is thus a promising manufacturing technology.

The origin of EV-producing cell lines also profoundly affects their intravesicular contents, and thus safety. Almost all mammalian cell types secrete exosomes, and many cell types, including stem cells, immune cells, and transformed cells, have been used to generate clinical-grade exosomes. EVs circulating in various biological fluids (serum, milk, urine, saliva, semen, etc.), originating from humans, other mammals, or plants, have been proposed as delivery or therapeutic/theranostic vehicles [126]. As incorporation of CRISPR/Cas components after EV formation and budding from a cell is currently unfeasible, in this review we only consider the cell lines and cell-based technologies that can be engineered for packaging CRISPR/Cas systems.

Most normal (non-transformed) cells typically have limited proliferation capacity [127]. Short-lived cell lines are poorly suitable for manufacturing EVs for clinical use. In contrast, transformed cell lines and stem cells represent a promising source for continuous generation of ready-to-use EVs. Most commonly, EVs are produced in transformed cell lines. The major advantages of these cells include rapid and almost indefinite cell expansion, potent production of exosomes, and the ease of generating stable engineered cell lines. The appealing properties of transformed and stem cell lines come with additional risks. EVs derived from cancer cells harbor numerous cancer-related molecules that can promote cell transformation, tumor growth, and cell invasion [128,129,130]. In addition, EVs derived from cancer cells can inhibit anti-tumor T-cell responses by inducing FasL, TRAIL, and PDL-2-related apoptosis [131] Carefully choosing EV-producing cell lines and strictly controlling their composition are instrumental for ensuring the safety of EV nanotherapeutics. Stem cells such as MSCs, although they do not undergo malignant transformation during long-term passaging [132], still accrue mutations [133]. The potential impact of such mutations on EV composition remains to be investigated.

A combination of properties, such as low immunogenicity, genetic stability, good proliferation potential, and a wealth of clinical safety data make MSCs a valuable source for manufacturing EVs. However, none of these properties of MSCs are perfect, and further improvements are required.

A recent study showed that repeated administration of EVs does not induce cellular toxicity [111], and EVs were safe and feasible in several phase I and II clinical trials in cancer patients [134]. However, even EVs produced from non-transformed cell lines may detrimentally affect certain cohorts of patients. For instance, MSC-produced EVs promote angiogenesis and transfer cargo that may potentially increase tumor growth and invasiveness; these events occurred in several models of tumor growth [135,136,137,138]. Several other studies reported conflicting results, showing anti-tumor and anti-angiogenic effects of MSC-derived EVs [139,140,141,142]. The conflicting results may be related to the origin of MSCs [143], culture conditions [144], and EV isolation methods [145].

The issue of up-scale production of MHC-derived EV is far from being solved. Generating MSCs from induced pluripotent stem cells (iPSCs) or embryonic stem cells (ESCs) devoid of MHCs, for instance by CRISPR/Cas cleavage of MHC loci [146], may become a solution for an endless source of clinical-grade MSCs. The capacity of cells to produce EVs with a consistent phenotype has to be better controlled for large-scale manufacturing of clinical-grade EVs. Many factors, including pH, shear force, and type of culturing (adherent, suspension, multilayer, bioreactors, 3D employing microcarriers, mass transfer), as well as the source of EVs, may alter EV characteristics and their therapeutic and targeting properties [147]. Contamination of EVs by serum constituents (if serum-free conditions are not used), mycoplasma, endotoxins, or bacteria, and batch-to-batch variation are other factors that may affect EV quality and their in vivo performance and production efficiency. The post-production pipeline is also important, as it may affect the quality of intravesicular cargo and the general biocompatibility and safety parameters of the derived EVs.

### 2.4. Biodistribution of EVs

Upon systemic administration, vesicle nanoparticles become undetectable in circulation after 4 h. Naturally occurring EVs are rapidly distributed to the major secretion organs [148]. Upon intravenous administration, unmodified EVs primarily localize to the liver, spleen, and gastrointestinal tract [149]. Radioactively labeled EVs were distributed into the liver (28% radioactivity), spleen (1.6%), and lungs (7%) [148]. Another study showed EV enrichment in the lungs, liver, spleen, and kidneys [150]. Intraperitoneally injected EVs accumulated in the liver, lungs, and spleen [151]. Accumulation of EVs in certain organs, such as spleen and liver, is commonly attributed to the circulating or sentinel lymphocytes and macrophages that retain and neutralize EVs [152,153].

It is well established that the size, shape, and surface charge of nanoparticles are important biophysical characteristics affecting the rate and type of their internalization by the recipient cells [89]. Serendipitously, Zhang et al. fractionated EVs produced from a single cell source into three subsets based on size and demonstrated that these populations exhibited distinct biodistribution profiles [154]. All types of EVs were uptaken by the liver (~84%), spleen (~14%), bone marrow (~1.6%) and, to a lesser extent, distributed into the lungs (~0.23%), kidneys (~0.08%), and lymph nodes (~0.07). Large EVs preferentially accumulated in the liver, bone, and lymph nodes, whereas smaller EVs were highly enriched in the liver [154]. EVs produced by many cell types have homing properties [155,156], i.e., preferential localization to sites of tissue damage [157] and inflammation [158]. Moreover, EVs originating from different cells appear to accumulate in distinct tissues and organs. This property can be used to increase the therapeutic capacity of EVs. Damage of EV membranes (by electroporation, freeze-thaw cycles, etc.) or degradation of EV surface markers (e.g., by protease digestion) alter EV accumulation in target organs may impair their ability to cross biological barriers [159,160,161]. For example, protease digestion of EV surface proteins reduced their accumulation in the lungs [162].

In general, the majority of wild-type exosomes or exosomes tailored with specific ligands either by covalent binding, genetic engineering, membrane integration, or encapsulation strategies were directed to the liver, lungs, spleen, kidneys, stomach, or bladder. In previous studies, EVs also localized to the thyroid, kidneys, brain, intestine, heart, muscle, bone marrow, blood cells, and pancreas, depending on the strategy utilized for surface engineering, the dose, cell source, and route of administration (reviewed in [163]). Enhanced uptake of EVs can be achieved using membrane surface engineering (cloaking approach, surface display) [164] or using a specific source of EVs for a certain disease [165]. Natural biodistribution may or may not be favorable for certain clinical applications. Several studies suggest that EVs are primarily internalized by the same cell types as those from which they originate [166]. EVs from a transformed HEK293T cell line, a common model system for biological experiments, accumulated in tumor tissues when injected in vivo, suggesting that these EVs can be harnessed for anti-cancer treatment [167]. The intrinsic property of EVs derived from ascites, hepatoma cells, and other various cancer cells to home to tumor tissues has been used to deliver anticancer drugs to treat glioblastoma, colorectal cancer, and hepatocellular carcinoma. Anticancer drugs packed into EVs from brain endothelial cells were successfully delivered across the blood-brain barrier to cure brain cancer [168]. In terms of infectious disease and in many types of cancer, EVs have been used as carriers of antigens to achieve therapeutic effect via induction of an immune response. However, EVs have also been provisionally utilized to deliver siRNAs against HCV into liver cells [169].

Ultimately, both the source of EV production and the engineered properties of EVs (so-called tailored or functionalized vesicles) affect their pharmacokinetics. Refining the surface characteristics of EVs by current and prospective molecular and cellular biology techniques may pave the way for generating highly specific and versatile delivery vehicles suitable for delivering gene-editing systems to treat virtually any human disease. Understanding additional factors aside from the cell source, integrity of EV membranes, and availability of tissue-specific ligands is important and may be highly beneficial for developing EV-based therapies.

## 3. Engineering the Surface of EVs for Improved and Targeted Delivery

The major challenge for in vivo applications of CRISPR/Cas is precise delivery of therapeutic complexes to the target sites in the body. Proper biodistribution is particularly important for gene therapy. An off-site delivery may have a direct toxic effect or increase editing at non-target genetic loci. A critical issue is a potential non-specific editing of germ cells, which may lead to mutations passed down to the offspring; besides the potential danger of developing new diseases, such a possibility also raises serious ethical concerns [170]. Therefore, robust and precise delivery of CRISPR/Cas RNPs with minimal off-target accumulation is mandatory for translating these gene-editing technologies to the clinic.

Both post-production physical/chemical modification and genetic engineering of the producer cells can be harnessed to add diverse functional ligands to the surface of EVs and ensure their accumulation in the tissues or organs of interest. As physical methods may damage EVs [159], the addition of functional ligands onto the surface of EVs using genetic engineering approaches and chemical methods appears a much more potent direction.

### 3.1. Genetic Engineering

The first study demonstrating the feasibility of manipulating EV tropism was published in 2005; Delcayre et al. used “exosome display technology” to embed targeting moieties into the surface of EVs by fusing them with the C1C2 domain of the lactadherin protein [171]. In another pioneering study, Alvarez-Erviti et al. (2011) generated siRNA-loaded exosomes with a genetically engineered surface [97]. The authors directly fused Lamp2b, a protein abundant on EV surface, with short peptides from rabies viral glycoprotein (RVG) or muscle-specific peptide (MSP), targeting EVs into the brain or muscle, correspondingly. Notably, injection of wild-type exosomes encapsulating siRNAs by nucleofection did not detectably decrease target gene expression in any organ analyzed. In contrast, RVG-enriched EVs carrying siRNAs significantly knocked down the target gene in several regions of the brain. RVG peptide, which specifically binds to the acetylcholine receptor, enabled EVs to target neurons, oligodendrocytes, and microglia. In contrast, MSP EVs demonstrated weak targeting capabilities in vivo, either due to low targeting ability of the MSP peptide or due to the high dilution of a single dose of MSP EVs across the highly abundant muscle tissue in the body. The milestones achieved in this study include not only the first indication that engineered EVs can traverse the blood-brain barrier to deliver cargo into the brain, but also that genetically engineered targeted EVs can escape non-specific sequestration in the liver, kidneys, and spleen.

A modification of this technology prompted design of EVs that selectively enter CD19^+^ B cells [172]. Surface display of integrin α6 with integrin subunits β1 and β4 targeted EVs to the lungs; expression of integrin β5 resulted in preferential accumulation of EVs in Kupffer cells in the liver; and abundance of surface integrin β4 increased recruitment of engineered EVs into endothelial cells in the brain [173]. Overexpression of CD63, a common tetraspanin marker of exosomes, determined EV enrichment in neuronal and glial cells [174]. In another study, tissue-specific targeting of EVs into the pancreas was achieved by exposing tetraspanin Tspan8 and integrin α4 on the EV surface [175].

In addition to EV surface proteins, EV lipid composition [176] and surface glycans [177,178,179] were shown to be important determinants of EV biodistribution and uptake. Tian et al [180]. targeted EVs into cancer cells by engineering Lamp2b protein fused with iRGD (CRGDKGPDC) peptide, allowing specific binding for integrin α5. Such engineered EVs delivered doxorubicin to cancer cells and inhibited their proliferation in vivo. Besides tumor tissues, however, engineered EVs also accumulated in the liver and spleen of mice. Overall, cancer-targeting EVs showed no evidence of general toxicity while effectively delivering chemotherapeutics to solid tumors. Similarly, Kim et al. [181] exploited the Lamp2b protein fused with cardiac-targeting peptide (CTP; APWHLSSWYSRT) to target HEK293-derived EVs into the heart. In vitro studies indicated increased uptake of CTP-modified EVs by cardiomyocytes although HEK293 cells were transfected by CTP-modified and unmodified EVs with similar efficiency. CTP-modified EVs injected into mouse tail veins were enriched in mouse hearts by 15% compared to unmodified EVs, whereas both types of EVs accumulated similarly in the liver and spleen. Identifying more robust cardiac-specific peptides or implementing other technologies, such as nanobody or multiplexed display, may achieve more beneficial parameters of EV delivery for heart diseases.

Developing EVs equipped with tissue-specific targeting ligands has a major limitation: it requires overexpression of an EV-specific ligand fused to an additional genetic construct. Overabundance of EV-specific proteins may have undesirable effects on EV-producing cells, ultimately affecting EV yield and composition. Second, certain ligands may compromise the stability of Lamp2b, resulting in premature degradation of fusion proteins. Indeed, Lamp2b requires post-translational modifications to maintain stability of the fused protein [182]. To overcome this problem, Zachary et al. constructed a set of proteins of interest (POI) fused with tetraspanins (CD9, CD63, CD81) for displaying them on the outer or inner surface of EVs [183]. CD63 is the most abundant tetraspanin on EV surface and can be utilized to robustly place POI on EVs [184]. Thus, fusing targeting moieties with tetraspanins is a viable and straightforward technology. How overexpressed tetraspanins impact the producer cells and EV composition is currently not clear.

Kooijmans et al. exploited the fact that EV membranes are highly enriched in glycosylphosphatidylinositol-(GPI) anchored proteins [185]. The authors hypothesized that cloning a GPI anchor signal peptide and fusing it to the POI may decorate EVs with targeting proteins. They programmed EVs to target tumor cells by displaying GPI-linked EGFR nanobodies. In a series of in vitro experiments with cell lines differentially expressing EGFR, the authors demonstrated that engineered EVs significantly associated with EGFR-enriched cells (A431) but showed poor binding to cell lines expressing low EGFR levels (HeLa and Neuro2A). Thus, not only can GPI fusions be a successful strategy for displaying ligands on the surface of EVs, but nanobodies targeting tissue-specific proteins may also dramatically improve EV accumulation in target tissues. Most recently, Cheng et al. devised the so-called synthetic multivalent antibody retargeted exosomes (SMART-Exos) by genetically engineering EV-producing cells to display two types of single-chain, variable fragment scFv antibodies on their surface (anti-CD3 to target T cells and anti-EGFR to target triple-negative breast cancer cells) [186]. The fusion partner for displaying the nanobodies was PDGFR, a transmembrane protein enriched in EVs. This study showed for the first time that EVs with dual surface nanobodies can target tumor cells and induce anti-tumor immunity by recruiting cytotoxic T cells to cancer cells.

### 3.2. Chemical Methods

Displaying EV surface proteins may be very reliable and reproducible, but requires genetic modification of EV-producing cells. This necessitates the generation of stable cell lines or transfection of large, manufacture-grade amounts of cells, which is either technically difficult or may preclude the production of easy-to-use EVs for targeting certain tissues/organs. Of more interest is the functionalization of EV surface by chemical methods, which enable rapid post-production targeting of EVs. Chemical approaches may rely either on co-incubation of EVs with EV-loading targeting peptides or covalent conjugation of ligands to the lipid or protein constituents of EVs via different linker groups, such as targeting transferrin receptors [187].

#### 3.2.1. Click-Chemistry

Wang et al. first proposed the use of an easily adaptable and highly versatile methodology to shape the exosome surface using copper-catalyzed azide alkyne cycloaddition, or so-called click-chemistry [188]. In this study, the authors first introduced azides into EV surface constituents by replacing the natural methionine in newly synthesized proteins with L-azidohomoalanine, an azide-bearing methionine analog. L-azidohomoalanine then served as an active site for conjugation through click-chemistry approaches. This technology can also be utilized to anchor targeting ligands to EV glycans and glycoproteins. Either L-azidohomoalanine or tetra-acetylated N-azidoacetyl-D-mannosamine, an azidosugar, are utilized by the cells during EV biogenesis and then exposed on the surface of EVs. Successful display of azide moieties on the EV surface was shown by incubating EVs with Cy3-conjugated dibenzobicyclooctyne, a far-red fluorescent dye used for visualizing azide biomolecules. Bio-orthogonal copper-free azide alkyne cycle addition was used to target cerebral vascular endothelial cells by attaching a targeting peptide to the surface of MSC-derived EVs [189]. In this case, click-chemistry enabled robust interaction of EVs with αvβ3 integrin, abundantly expressed in the ischemic regions of the brain, suppressing the local inflammatory response, and reducing cellular apoptosis by co-delivering curcumin in mice. Smyth and co-authors functionalized EVs with alkene groups to conjugate them with fluorescent molecules using click-chemistry [190]. Assembling targeting modalities on the surface of EVs using this method may prove useful for re-distribution of EVs in vivo, given that this modification neither impairs EV characteristics nor is toxic to the cells.

#### 3.2.2. Painting EVs with Targeting Peptides

In 2018, Gao et al. performed a high-throughput screening for peptides specifically binding to the second extracellular loop of tetraspanin CD63 and identified peptides CP05 and CP07, which can be conjugated onto EVs with different tissue-targeting moieties in a process termed painting [191]. Painting CP05-conjugated EVs with muscle-targeting (M12), brain-targeting (RVG), or hepaptocellular carcinoma-targeting (SP94) peptides induced redistribution of exosomes into the muscle tissue, brain, or tumors, correspondingly. Notably, M12-modified EVs accumulated in the liver and, sporadically, the spleen, in addition to efficient amassing in various muscles. RVG indeed directed EVs into the brain, substantially reducing accumulation in the liver, but RVG-painted EVs were also detected in the spleen and kidneys. SP94 peptide enriched EVs in tumor tissues, but EVs were also distributed in the liver, spleen, lungs, and kidneys. The important advantage of these visualizing peptides is that when conjugated with several targeting ligands, they can ensure dual targeting of EVs. Gao et al. incubated EVs with CP05-FITC and CP05-rhodamine dyes simultaneously and observed a strong cumulative signal on the same modified EVs. Besides this CD63-specific peptide approach, EVs can be conjugated with streptavidin. Coupling streptavidin-coated EVs with biotinylated tissue-specific antibodies or nanobodies may be a useful approach to rapidly decorate EVs with targeting modalities and provide desirable re-distribution in vivo.

#### 3.2.3. Displaying Targeting Ligands by Co-Incubation with Liposomes or Synthetic Peptides

Recently, several regimens have been tested to produce hybrid EVs upon fusion with synthetic lipid nanoparticles. This membrane-engineering approach is based on freeze-thaw-induced fusion of EV membranes with lipids of synthetic nanoparticles [192]. The drawback is related to the increased toxicity of mixed nanoparticles, as well as potentially reduced ability to cross biological barriers (e.g., endothelial barrier or endolysosomal degradation) due to their increased size and altered composition [193]. Embedding fusogenic peptides (e.g., GALA) may be sufficient to circumvent endolysosomal degradation. Distribution and uptake of EVs may be also altered by exposing different peptides on the EV surface, such as arginine-rich micropinocytosis-inducing peptide [101]; adding cell-penetrating peptides; activating receptors on the surface of EVs [194]; conjugating specific targeting moieties [195]; displaying vesicular stomatitis virus (VSV) G protein (VSV-G) for increased tropism [196]; or displaying targeting nanobodies/antibodies as highlighted above. Importantly, display of certain moieties on the EV surface can improve the release of active payloads into target cells, as was shown for surface antibodies that shifted EV uptake from caveolae-mediated endocytosis to micropinocytosis, thereby avoiding the endolysosomal pathway and improving efficacy of the intracellular cargo delivery [197]. Directing EVs into pre-determined locations in the body was also achieved by utilizing aptamers, nucleic acids specifically recognizing target molecules. In one study, RNA and DNA aptamers directed siRNA-loaded EVs into the cancer cells, reducing tumor growth in mouse models [198,199].

## 4. Packaging CRISPR/Cas Protein and RNA Components into EVs

A specific difficulty in delivering gene editing is the need to co-package and protect two kinds of molecules, protein, and RNA. This task multiplies all hurdles associated with a single cargo type, and a plethora of trials resulted in the emergence of a variety of methods for both types of payload. Existing molecular tools and mechanisms for loading Cas proteins into EVs are listed in Table 1.

### 4.1. Cas Protein Packaging

#### 4.1.1. WW-Ndfip1 Interaction and Post-Translational Modifications

Proteins that contain so-called late domain-motif (L-domain), for example, Ndfip1, are recruited into EVs with the aid of the intracellular sorting machinery [208]. PPxY L-domain motifs of Ndfip1 bind WW domains of Nedd4 protein and contribute to its monoubiquitination and packaging into EVs. This property of the Ndfip1 protein was exploited by Sterzenbach et al., who designed a Cre recombinase fused with a synthetic WW peptide, thus enabling packaging into EVs by Ndfip1-overexpressing cells [86]. Notably, the Cre recombinase was functional in target cells. However, overexpression of Ndfip1 could be toxic for the cells. Therefore, it is necessary to design engineered WW-Ndfip1 interactions that preserve EV packaging but reduce or eliminate cell toxicity.

Ndfip1-WW interaction mediates Cre recombinase transfer into EVs by monoubiquitination (Figure 1A). Previous studies demonstrated that fusing POI with ubiquitin motifs is sufficient for their enrichment in EVs [209]. For instance, monoubiquitination of syntaxin 3 or MHC-II effectively packaged these proteins into EVs [210,211]. Other post-translational modifications of proteins are also associated with EV packaging: myristoylation an [202,212]. Whether POI with such tags can be released from the EV membrane and remain functional in target cells remains to be investigated. This method has not been employed for Cas protein delivery, but given the universal mechanism this method relies on, it can potentially be utilized for Cas packaging into EVs.

#### 4.1.2. Arrestin-Domain Containing Protein 1 (ARRDC1)-Mediated EVs

While the methods described above mostly used exosomes as EVs, Wang et al. utilized a different type of EVs called arrestin domain containing protein 1- (ARRDC1) mediated microvesicles (ARMMs) [204]. These vesicles do not rely on ESCRT machinery and fuse directly with target cells, avoiding the transfer of enclosed cargo into the endolysosomal compartment, potentially resulting in more efficient cargo release compared to exosome-based delivery vehicles. Wang et al. took advantage of the ARRDC1 PPxY motifs interacting with WW domains of the ITCH protein. WW-Cas9 proteins were co-expressed with ARRDC1 in ARRM-producing cells and robustly incorporated into ARMMs. This method of packaging is very simple and should have little impact on the functioning or nuclear localization of the Cas9 protein in target cells. Production of ARMMs in non-transformed cell lines and the distribution and homing of ARMMs in vivo are issues that need to be addressed in further studies.

#### 4.1.3. Nanoblades

Another type of delivery system that takes advantage of artificial EVs generated by HIV Gag proteins has been named Nanoblades^211^. Nanoblade technology takes advantage of the viral structural Gag polyprotein which multimerizes at the cell membrane in the presence of retroviral fusogenic viral envelope and induces the release of virus-like particles. Fusion of Gag murine leukemia virus-like particles (Gag-Pro-Pol) with Cas9 produced small (150 nm) virus-like nanoparticles that successfully delivered Cas protein into target cells (Figure 1B). In this setting, expressed Gag-Pro-Pol protein-induced assembly of virus-like particles containing a fusion Gag-Cas9 protein. Gag and Cas9 protein are separated by a proteolytic site which is cleaved by the MLV protease, releasing free Cas9 protein into the lumina of nanoparticles. Importantly, in the above study, Nanoblades carried only minor amounts of carry-over intracellular proteins and RNA into receptor cells, as demonstrated by the almost complete absence of endogenously overexpressed firefly luciferase. Intriguingly, Nanoblades successfully packaged sgRNA. Packaging of sgRNA into Nanoblades was strongly dependent on the interaction of Gag with Cas9. Characteristics of nanoblade systems are provided in Table 1 and Table 2. The number of Cas proteins packaged into a nanoblade vesicle needs to be defined in different cell types. Furthermore, the liberation of the Cas protein from Gag by MLV protease may pose a risk of non-specific proteolytic cleavage of Cas protein or cellular proteins as suggested previously [201].

#### 4.1.4. VEsiCas

The ability of VSV-G envelope glycoprotein to generate vesicles prompted Montagna et al. to evaluate VSV-G-formed nanoparticles for delivering Cas9 proteins and sgRNAs [206]. This method is based on transfecting HEK293T cells with plasmids expressing Cas9, VSV-G, and sgRNA, and passive loading of CRISPR/Cas9 components into the newly formed vesicles. Notably, in this approach, CRISPR/Cas9-components are not physically fused or pushed to interact with vesicle membranes as in the aforementioned approaches. In the end, VSV-G generated vesicle nanoparticles that incorporated both Cas9 protein and sgRNAs, but the amount of Cas9 comprised only 1.5–2% of the total protein content.

#### 4.1.5. Gesicle System

A strategy similar to VSV-G nanoparticles was used in the “gesicle” system [208]. Nanoparticles were engineered with CherryPicker Red membrane-associated protein fused with a dimerization DmrA domain that interacts with the DmrC domain fused to Cas9 protein. DmrA and DmrC domains interact upon addition of an A/C heterodimerizer molecule, resulting in stimulus-induced dimerization and packaging of Cas9 proteins. The authors noted several evident drawbacks, such as the short half-life of Cas9 protein in gesicles (less than 24 h), possibly due to proteolytic cleavage of Cas9 inside gesicles, and the low proportion of gesicles harboring RNPs (<1%). An unfavorable observation was that gesicle formation increased upon transfection of producing cells, indicating that the gesicles may represent cellular waste and by-products ejected by the cells. Low packaging capacity and yet-undefined composition of gesicles limit their use. However, optimizing this platform with sgRNA-packaging devices, as well as enhancing Cas9 packaging, may put gesicles on par with such platforms as ARMMs. The overview of CRISPR/Cas-packaging platforms with a critical assessment of their pros and cons is provided in Table 2.

### 4.2. sgRNA Loading into EVs

After the historic discovery that different RNA types are present inside EVs [213,214], the specific loading mechanisms for different types of RNA have been investigated. A plethora of studies demonstrated that even very abundant intracellular RNAs may be virtually absent in released EVs [215]. Thus, successful sorting of RNAs into EVs requires an exhaustive understanding of the molecular mechanisms driving specific RNAs into the EV compartment. These sorting mechanisms are not completely understood and hard to control. The most recent review of the sorting mechanisms of different types of RNA into EVs is provided by O’Brien with co-authors [215].

Targeting sgRNAs must be co-packaged along with Cas proteins for successful delivery of CRISPR/Cas RNPs via secreted EVs. General classification of EV loading techniques distinguishes endogenous loading (transfer of endogenously transcribed RNA into EVs) (provisional technologies are summarized in Figure 2) and exogenous loading (delivery of RNA into isolated vesicles) (see Table 3). Endogenous loading has the major disadvantage that sgRNA is produced in the same cells as the Cas protein; thus, EV-producing cells can be modified by the CRISPR/Cas system, possibly impairing their characteristics. Moreover, loading of both Cas protein and sgRNA into EVs should be controlled and validated at the same time to ensure optimal production and packaging of each component inside the cells. Nevertheless, simultaneous co-packaging of Cas protein and sgRNA into a single nanoparticle is advantageous, as the two components immediately generate a reactogenic complex inside EVs and can exhibit high gene editing efficacy upon delivery into target cells. In turn, exogenous loading mechanisms can deliver any type of sgRNAs (even those that may dramatically affect the viability of EV-producing cells) in large quantities into Cas-containing EVs, and are very amenable to large-scale production of EVs targeting different tissues for editing with various sgRNAs. However, commonly used techniques for exogenous loading damage EVs, affecting their physical and biological properties.

#### 4.2.1. Endogenous Packaging of RNAs for CRISPR/Cas Applications

##### The Use of EV-Associated Motifs

Recent studies have revealed the role of EV-associated motifs or “zipcodes” that ensure preferential egress of RNAs inside EVs. One of the most fascinating discoveries of RNA packaging into EVs was made by identifying that EVs from various cell lines are strongly enriched in miR451. This micro RNA has a unique biogenesis mechanism that does not rely on the common cleavage of pre-miRNA stem-loop structure by the dicer enzyme but instead harbors a unique short stem-loop (51 nt instead of 70–120 nts in other pre-miRNAs) [233,234]. Replacing the long pre-miRNA stem-loop with the short miR451 loop or its modified structural mimic results in over 58-fold to 70,000-fold enrichment of engineered miRNAs in exosomes without affecting their activity and specificity [216]. This technology is broadly applicable as it utilizes short motifs that can be used to engineer RNAs for relaying into EVs, but its utility for sgRNAs loading needs to be experimentally verified (Figure 2A). Cell type specificity may also be important, as enrichment rate varied among cell lines, and EVs produced by embryonic stem cells contained almost no miR451 transcripts.

On the other hand, several studies identified exosome secretion motifs (EXOmotifs [217,235,236]) as well as annexin A2-interacting motifs targeting RNA molecules into EVs [220]. These studies proved that RNA is not packaged indiscriminately into EVs, but instead relies on an intricately regulated intracellular sorting machinery. These sorting motifs are summarized in Table 3. RNA carrying these motifs are bound by intracellular RNA-binding proteins (RBPs) [237,238] and enriched in EVs; mutations in these motifs impair EV trafficking [237]. However, active loading of specific sgRNAs using these moieties is hard to achieve, as they require the presence of several targeting sequences forming a complex secondary structure. Differences in the RBP levels may result in cell type-specific differences in RNA sorting. These moieties in general poorly enrich RNA in EVs and thus have not been used for RNA/sgRNA packaging yet. Identifying the mechanisms and key players involved in selective loading of endogenously produced RNAs/sgRNAs is particularly important for EV-mediated sgRNA delivery.

##### Overexpression Strategy

Overexpression of RNA or protein inside a producer cell does not guarantee its effective packaging into EVs [202]. Yet simply overexpressing CRISPR/Cas constituents has been independently shown by two laboratories to package both Cas9 protein and sgRNAs into EVs [203,239]. Lainšček et al. coined the term GEDEX for stochastic packaging of CRISPR/Cas9 components into EVs produced by transformed HEK293 cells [203]. The authors attempted to increase the packaging efficiency of CRISPR/Cas9 components by co-expressing CD9 or neutral sphingomyelinase-2, which both play an important role in EV biogenesis (provisional technology in Figure 2B), or by designing a Cas9 protein with a C-terminal CAAX box farnesylation motif. However, these efforts did not significantly increase CRISPR/Cas loading. Chen et al. demonstrated as well that CRISPR/Cas9 RNPs can be co-packaged into exosomes produced by different transformed cell lines (HEK293/T, HepAD38, HeLa, and Huh-7) [239]. This stochastic co-packaging is simple and does not require additional modification of the producer cells, but the ability of non-transformed cell lines to package CRISPR/Cas components is unknown. Stochastic RNP loading into EVs also seems inefficient, as Gee et al. did not observe significant CRISPR/Cas RNPs packaging and on-target gene editing by EVs produced in HEK293T cells [201].

##### Light-Inducible MS2-MCP Interaction of sgRNAs with EVs-Constitutive Proteins

An alternative RNA-packaging approach based on reversible protein-protein interaction was exploited by Huang and co-authors (Figure 2C) [222]. This group used a blue light-mediated reversible CIBN-CYR2 interaction to package RNA. CIBN was fused with EV-associated motif and MS2 bacteriophage coat protein (MCP) that interacted with RNA-aptamer MS2-RNA upon blue light illumination. Induced blue light illumination resulted in enrichment of target RNA in EVs, whereas absence of light led to dissociation of MS2 and MCP and release of MS2-RNA into EV lumen. Injection of EVs allowed delivery of functional MS2-RNA into target cells. This technology has not been adopted for sgRNA, but seems to be a viable approach. The major confounding aspect of this platform is the requirement of blue light illumination, which is notorious for inducing apoptosis and cell death [240]. Prolonged exposure to blue light may result in death of EV-producing cells, drop of EV production, or result in potentially hazardous substances, such as reactive oxygen species, which may be packaged into EVs [241].

##### Synthetic Constructs for Loading sgRNAs into EVs

Gee et al. designed a synthetic system for actively loading both Cas proteins and sgRNAs into EVs (Figure 2D). The authors created a construct harboring a Psi (Ψ+) sequence that interacts with the endogenously produced HIV Gag protein to package sgRNA into exosomes [201]. Additionally, sgRNA was flanked by the self-cleaving ribozymes hammerhead (HH) and hepatitis D virus ribozyme (HDV). sgRNAs were actively loaded into EVs and released free of packaging signals inside the target cells. This method is at least four times more efficient at loading sgRNAs than stochastic sgRNA incorporation and can be used with any sgRNA and producer cell lines.

##### EXOtic RNA-Packaging Device

Another approach for packaging sgRNA into EVs was inspired by synthetic biology and utilized a two-component system consisting of a CD63 fused with L7Ae archaeal ribosomal protein interacting with a modified RNA with a C/D box inserted in the 3′ UTR of the reporter gene. Connexin 43, a protein enriched in exosomes [242], was used as a cytosolic delivery helper enhancing RNA sorting into exosomes. This RNA packaging system was named exosomal transfer into cells (EXOtic) [221]. Notably, while EXOtic also enhanced exosome biogenesis and secretion by distinct mechanisms, successful transfer of RNA into EVs was not converted into increased efficacy in target cells. This is likely because RNA cannot be released from EVs (the L7Ae-RNA interaction may be irreversible), or because exosomes with incorporated RNA enter the endolysosomal pathway, resulting in degradation of cargo [242]. Similar problems occur with the targeted and modular EV loading platform (TAMEL), which utilizes the fusion of an EV-enriched protein with an RNA-binding bacteriophage MS2 protein dimer interacting with MCP-RNA [222]. While this method provided robust RNA packaging into EVs, RNA delivery into the recipient cells did not result in protein translation.

#### 4.2.2. Exogenous Loading Approaches

An alternative RNA loading approach is post-production insertion into EVs with pre-packaged Cas proteins. It appears more plausible than endogenous delivery, as EV-producing cells do not undergo gene editing and thus retain their original phenotype. An important benefit of exogenous sgRNA packaging is that additional modification of the EV-producing cells is not needed. Moreover, exogenous encapsulation potentially offers higher versatility in terms of packaging well-measured, controllable, and tunable amounts of sgRNAs per vesicle, and even loading modified/synthetic sgRNAs, as opposed to endogenous methods that rely on cellular machinery.

Post-production encapsulation of RNA is possible by co-incubating EVs with RNA in certain conditions. A unique complication of sgRNA delivery compared to small RNA/mRNA are higher quality requirements. Several lines of evidence indicate that sgRNA may induce a strong innate intracellular immune response, resulting in substantial (>80%) death of transfected cells; this seems to be due to a cytotoxic 5′-triphosphate group that can be recognized by RIG-I RNA-sensor [18]. Treating in vitro transcribed sgRNA with phosphatases (such as calf intestinal phosphatase) helps to avoid this innate intracellular immune recognition and achieve effective genome editing. In vitro production of sgRNA using phage RNA polymerases (T7 or SP6) is admittedly the easiest method, suitable for clinical applications and manufacturing. Chemical synthesis is a less affordable method, but although it produces lower amounts of sgRNA, it installs a delicate control over sgRNA properties with a single-nucleotide resolution. Chemical synthesis of sgRNA lacking 5′- triphosphate group prevents sgRNA recognition by innate intracellular immunity, eliminating the need for phosphatase treatment. Chemical modifications of 3′ and 5′ ends of sgRNAs (2′-O-methyl, 2′-O-methyl 3′-phosphorothioate, or 2′-O-methyl 3′-thioPACE) at three terminal nucleotides not only substantially increased on-target indel frequencies (by >60–80% for 2′-O-methyl 3′-phosphorothioate or 2′-O-methyl 3′-thioPACE), but also increased the specificity of CRISPR/Cas, reducing undesirable gene editing at off-target loci to near-background levels [243]. Similarly, introducing 2′-O-methyl-3′-phosphonoacetate at select sgRNA sites dramatically improved sgRNA specificity while retaining gene editing activity [244]. Chemical modifications lower the free-energy barrier between the formation and dissociation of the complex between CRISPR/Cas RNP complex and the target DNA, shifting the thermodynamic barriers in favor of more accurate gene editing activity. Most recently, Taemaitree et al. coupled the bimolecular guide RNA system (hybridization of 42-mer crRNA with fixed 80-mer tracrRNA) with chemical modification of the crRNA/tracrRNA nucleotides using untemplated copper-catalyzed azide-alkyne cycloaddition chemistry to generate high-purity chemically modified sgRNAs in a cost-effective manner [245]. Indeed, chemical alterations in sgRNA compositions can increase gene editing activity and specificity of CRISPR/Cas systems if embedded in certain regions of sgRNAs. Further advances in chemical modification of sgRNAs and synthetic sgRNA production will pave the way for manufacturing high yields of high-purity sgRNAs with enhanced on-target and off-target properties.

Similar to the issues related to protein loading into EVs, exogenous encapsulation techniques should not damage EV integrity, given the potential rapid degradation of vesicular RNAs by RNases present in the blood and tissues. Harsh methods that rely on breaking and rebuilding EV membranes (e.g., freezing-thawing) may not only impair the natural or targeted homing properties of EVs but also deteriorate the quality of packaged proteins, including Cas. Potential destabilization of Cas proteins and disrupted integrity of EV membranes are the major challenges for exogenous encapsulation techniques. Additionally, changing the size, shape, and charge of EVs by physical disruption methods may substantially impair their ability to cross biological barriers and reach target sites in vivo. A number of currently available techniques are briefly introduced below.

##### Hydrophobically Modified RNAs

A different approach exploits hydrophobic moieties-mediated loading of RNAs into EVs. Several studies described a number of reproducible and scalable strategies for transferring chemically modified RNAs (mostly siRNAs) into isolated EVs. In particular, a group led by Anastasia Khvorova demonstrated that modifying miRNA with docosanoic acid, cholesterol, or tocopheryl succinate led to the packaging of thousands of miRNAs into EVs [224,225]. Packaging efficiency and stability of packaged RNA were affected by siRNA chemical modification patterns, position of the hydrophobic moiety (5′ or 3′ strand), and the linker used to attach the moiety to siRNA. The degree of hydrophobicity is of paramount importance for successfully enriching EVs. The stability of RNAs is mostly affected by modifications responsible for phosphatase resistance and, cumulatively, for nuclease resistance. Most importantly, co-incubation of chemically modified RNAs with EVs resulted in both surface-bound and luminal RNAs, though which RNAs are functional is yet unclear. Extensive chemical modification and cell type-specific optimization of chemically modified RNA may be required. The major limitation of this approach is that RNA may be poorly released from the EV surface, limiting its functional effect, especially in terms of CRISPR/Cas applications. While this method provides the packaging of many thousands of RNA molecules per vesicle, the efficiency remains fairly low (~43% of EVs). The utility of this method for sgRNA loading into EVs is also not yet defined.

##### Fusion with RNA-Loaded Liposomes

Incubating EVs with nucleic acid- (NA) loaded liposomes has emerged as an efficient transfer technique [227]. EVs are incubated with liposomes for 12 h at 37 °C. Similarities in the lipid composition of EVs and liposomes lead to fusion between their membranes, generating hybrid structures with different characteristics (e.g., charge and size distribution). Importantly, the majority of exosomes were successfully fused with liposomes and contained NAs. This approach was originally used to load DNA into EVs, but may be implemented to deliver RNA as well. Whether the yield of packaged RNA or pre-packaged Cas proteins may be reduced upon long-term incubation at 37 °C due to degradation remains to be determined. Moreover, liposomes exhibit toxicity in in vitro and in vivo studies. While EVs are highly biocompatible and do not affect cell viability, liposome/EV hybrids have both biocompatible and toxic components. Indeed, hybrid EVs were toxic to MSCs. The authors concluded that specific modification of liposomes to reduce toxicity will be necessary for clinical application of hybrid nanoparticles. Several additional aspects also need to be defined, such as the circulation time and biodistribution of hybrid nanoparticles, their ability to overcome biological barriers, and compatibility with target cells in vivo [226]. Despite a number of unresolved issues, the idea of loading EVs by mixing with liposomes is very appealing.

##### Physical Methods

A number of physical methods have been used to incorporate RNA into EVs, including electroporation [159], sonication [246], heat shock [229], pH gradient modification [231], freezing-thawing [228], and extrusion [231]. These methods destabilize EV membranes and allow incorporation of RNA by passive loading. The majority of these methods use very harsh conditions that not only significantly affect EV composition (by generating large holes in the membrane, causing aggregation, fusion of EVs, and resulting in loss of EV markers), but may also destroy the cargo. Electroporation, for example, generates holes of 5–10 nm in EVs, impairing their drug delivery properties. On the other hand, pH gradient modification and freezing-thawing reduce overall protein content and induce the loss of EV surface markers and cargo. Thus, these damaging methods have limited applicability for cargo packaging.

Although the methods for endogenous and exogenous RNA delivery into EVs are under development and many approaches for EV-mediated sgRNA transfer are currently available, many issues persist regarding biosafety, cargo and EV stability, release, and delivery of functional sgRNAs into the nuclei.

## 5. Conclusions

There is a growing need for a molecular vehicle that can successfully load and deliver CRISPR/Cas RNPs into target tissues. Synthetic delivery vehicles are being developed but so far have been only moderately successful. EVs are ideal candidates for a universal biological platform to produce ready-to-use, programmable, and highly biocompatible CRISPR therapeutics. Using EVs in the CRISPR/Cas research and, ultimately, in the clinic, demands novel, advanced techniques for protein/RNA loading, surface engineering, and manufacturing. Safety of CRISPR/Cas systems and EVs also need to be tested extensively for every particular application.

## Figures and Tables

**Figure 1 ijms-21-07362-f001:**
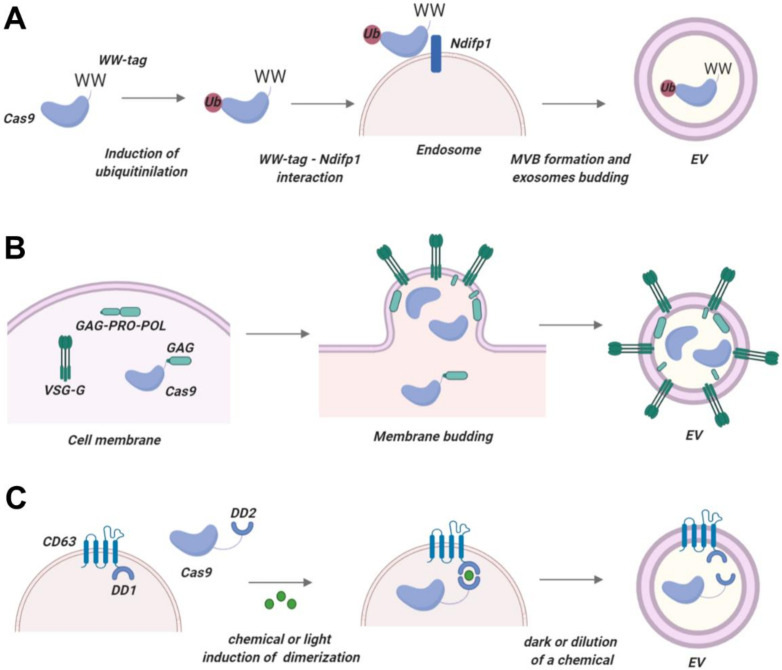
Packaging Cas proteins into EVs. (**A**) Provisional technology based on WW-Ndfip1 interaction. Cas protein with a WW tag can be expressed intracellularly together with Ndfip1. Overexpressed Ndfip1 mediates ubiquitination of Cas-WW and promotes its loading into EVs. (**B**) Nanoblade technology. Virus-like particles are generated by Cas protein fused with HIV Gag and co-expressed with Gag-Pro-Pol protein. Resulting EVs exhibit minor carry-over of cytosolic matter and effectively enter target cells. (**C**) Provisional technology based on the fusion of a constitutive EV membrane protein (e.g., CD63) with a dimerization domain and the use of a hybrid Cas protein with another dimerization domain. Upon signal (light or a chemical molecule), domains dimerize and Cas protein is recruited into EVs. Post-production, the signal is removed, and the Cas protein is released into the EV lumina. This picture was created in BioRender.

**Figure 2 ijms-21-07362-f002:**
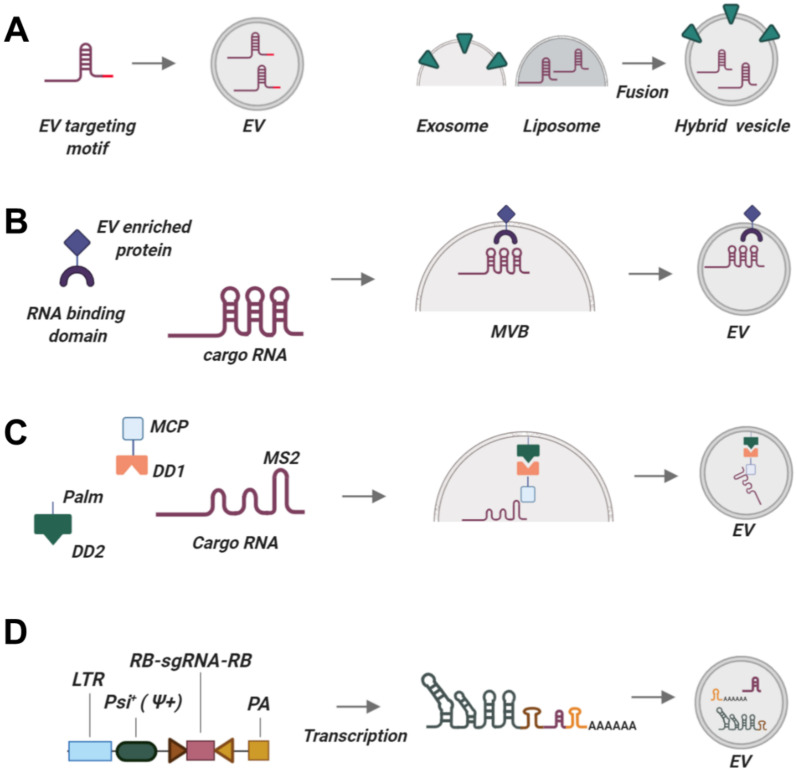
Existing and prospective technologies for tagging sgRNAs to be packaged into EVs. (**A**) Constructing synthetic RNA chimeras with EV-targeting motifs may enrich these RNAs in EVs. (**B**) Using EV-enriched proteins coupled with RNA-binding domains. (**C**) Packaging based on the interaction of MCP protein with an MS2 aptamer introduced into cargo RNA. Palm signal localizes to the membrane of EVs together with dimerization domain DD2. MCP is fused to dimerization domain DD1. Cargo RNA interacts with MCP via MS2 aptamer. Upon incoming signal (light or a small chemical), DD1 and DD2 dimerize, bringing together all three components so that cargo RNA is packaged into EVs. After EVs are produced, DD1 and DD2 dissociate, releasing cargo RNA into the lumina of EVs. (**D**) sgRNA packaging device, a part of the NanoMEDIC platform. A long RNA is encoded intracellularly, comprising the Psi+ EV-localization signal and two ribozymes. Upon loading into EVs, the long construct is self-cleaved by ribozymes, releasing the sgRNA with no additional RNA sequences. Abbreviations are explained in the text. This picture was created in BioRender.

**Table 1 ijms-21-07362-t001:** Molecular tools and mechanisms for packaging Cas proteins into EVs.

Name	Type of Cas Package	Cas Packaging System	Type of sgRNA Package	sgRNA Packaging System	Vesicle Producing Cell Line	Number of Cas:sgRNA RNP Complexes Per Vesicle	Notes
EXPLORs [200]	Optogenetic dimerization system	CRY2 interacting with CIB1 module via blue light illumination and transient docking of CRY2-POI proteins to EVs (schematically depicted in Figure 1C)	NA	NA	HEK293T	Unknown	Induced by blue light illumination
NanoMEDIC [201]	Chemical ligand-dependent dimerization	-FRB-SpCas9:FKBP12--Gag^HIV^ (schematically depicted in Figure 1C)	Packaging signal	HIV Ψ packaging signal with HH and HDV self-cleaving ribozymes	HEK293T (adherent and in suspension)	3,5-7,9	Production in xeno-free conditions is 30% less efficient
Tags for post-translational modification [202]	Post-translational modification	-Ubiquitination -Myristoylation -Palmitoylation	NA	NA	Cancer cells	-Not specified -May be cell type-specific	-Release and functionality of POI in target cells is unclear -Efficiency is unclear
Genome editing with designed extracellular vesicles (GEDEX) or stochastic packaging [203]	Stochastic	Overexpression	Stochastic	Overexpression	-HEK293 -HepAD38 -HeLa -Huh7	10 µg of EVs contain 100 ng of Cas9 protein	Tested in vivo and in vitro
WW-Ndfip1 interaction [87]	Ubiquitination of the target protein	Fusion WW domain linked to POI	NA	NA	MEFs	Unknown	-Ndfip1 overexpression is required for packaging -Ndfip1 is toxic to cells
Arrestin domain containing protein 1- (ARRDC1) mediated microvesicles (ARMMs) [204]	Fusion of Cas9 with 2–4 ITCH domains	ARRDC1:WW-Cas9 (ITCH WW domains)	Packaging signal	ARRDC1-Tat: TAR-RNA co-transfection	HEK293T	540 protein molecules	-Delivers cargo to many organs in vivo -Vesicle targeting and tissue specificity need to be tested
NanoBlades [205]	Gag^MLV^ fusion	SpCas9- Gag^MLV^ supplemented by Gag-Pol^MLV^	Unclear, depends on Cas9-Gag interaction	Depends both on interaction with Cas9 and Gag proteins, but is not elucidated	Adherent HEK293T	Unknown	NanoBlades are shed vesicles with unknown characteristics
VEsiCas [206]	Stochastic incorporation with VSV-G assistance	VSV-G-assisted accumulation at cell periphery and vesicle packaging	Transcription of sgRNAs in the cytoplasm	T 7 RNA Pol-driven transcription. HDV ribozymes between the sgRNA and T7 terminator generate mature sgRNAs with unmodified 3′-constant regions	Adherent HEK293T	1.5–2% of the total protein content of VEsiCas	VEsiCas are shed vesicles with unknown composition
Gesicle system [207]	Chemical-induced incorporation	-CherryPicker membrane-anchoring DmrA proteins associate with DmrC domain of Cas9-DmrC via A/C heterodimerizer molecule	Stochastic or mediated by interaction with Cas9	NA	Adherent HEK293FT	<1% of gesicles contain Cas9:sgRNA RNPs	-Very inefficient packaging -sgRNA package not enforced -Cas9 half-life reduced

**Table 2 ijms-21-07362-t002:** Characteristics of molecular tools and mechanisms for loading Cas proteins into EVs.

Name	Advantages	Drawbacks	Prospects
EXPLORs [201]	-Highly efficient -Utilize reversible protein-protein interaction modules -Transient protein docking into EVs	-Have not been used for CRISPR/Cas9 -sgRNA delivery is not addressed -Blue light is toxic to the cells	-Can be coupled with other light-induced dimerization (LID) or chemically-induced dimerization (CID) systems for sgRNA packaging -Cycles of blue light may be less toxic to producer cells
NanoMEDIC [202]	-Very first demonstration of successful exosome engineering for packaging and delivering CRISPR/Cas9 -Very efficient packaging of both Cas9 and sgRNA-Very efficient in vitro and in vivo genome editing -Proven activity in vivo-Cleared in vivo within 3 days -Scalable system in chemically-defined media with suspension cell culture	-Use HEK293T, a transformed cell line -Transformed cell lines produce exosomes with pro-oncogenic properties -Use rapamycin, an immunosuppressive drug with a number of potential adverse effects, to induce dimerization of domains. Rapamycin may potentially be packaged into EVs or alter exosome composition -Tissue-specific targeting upon systemic delivery has not been investigated -Reliance on HIV-1 Tat/Gag to drive sgRNA expression imposes the risks of toxicity both to producer cell lines and target cells -HIV-1 Tat/Gag may alter exosome composition -Co-produces Cas9 and sgRNA in the same cell	-Can be potentially expanded to clinically relevant EV-producing cell lines -Packaging of rapamycin into EVs and its effects on exosomes still needs to be defined -Any type of CRISPR/Cas system can be packaged
Tags for post-translational modification [203]	-Simple and feasible even for large proteins	-Have not been used for CRISPR/Cas9 -Most likely cell type-specific -Efficiency is unclear-Functionality in target cells unclear	-Simple and feasible -Applicability for Cas proteins needs to be defined
GEDEX or stochastic packaging [204]	-Very first demonstration of CRISPR/Cas9 RNP stochastic packaging into exosomes -Packaging of both Cas and sgRNAs -Efficient in vitro and in vivo genome editing -Scalable	-Utilize transformed cell lines -Transformed cell lines produce exosomes with pro-oncogenic properties -Tissue-specific targeting upon systemic delivery has not been investigated -Co-produce Cas9 and sgRNA in the same cell	-Very simple (overexpression of CRISPR/Cas components) -Any type of CRISPR/Cas system can be packaged
WW-Ndfip1 interaction [87]	-Efficiently delivers Cre-recombinase to target cells -Tested in vivo -Very simple (very short fusion peptides)	-Has not been used for CRISPR/Cas9 -Utilize mouse cells; not studied in human cells -Not studied with CRISPR/Cas packaging -Do not contribute to sgRNA packaging -Overexpressed Ndfip1 is required -Ndfip1 is toxic to producer cells -Ndfip1 interacts with numerous pro-oncogenic and pro-apoptotic factors	-Ndfip1 is toxic to producer cells -Ndfip1-WW interaction needs to be rationally engineered
ARMMs [205]	-Simple loading of protein and RNA cargo into vesicles -Efficient packaging of CRISPR/Cas RNPs -Efficient genome editing -Scalable -ARMMs may enter cells by direct fusion -Cargo bypasses endolysosomal pathway	-Use transformed cell lines -Transformed cell lines may produce vesicles with pro-oncogenic properties -Tissue-specific targeting upon systemic delivery has not been investigated -Co-produces Cas9 and sgRNA in the same cell	-Very simple packaging -Effects of ARRDC1 expression on producer cells and vesicle composition need to be addressed -Benefits of ARMMs over exosomes in terms of scalability and production need to be addressed
Nanoblades [206]	-Very limited carry-over of cellular proteins or overexpressed RNAs -Can potentially be produced from non-transformed cell lines -Have been combined with BaEV and VSV-G for improved delivery -Tested in vivo -Complex homologous DNA templates to generate knock-ins	-Carry-over of cellular RNAs (including those with pro-oncogenic potential) has not been investigated -Virus-like particles (viral origin) with membrane-associated proteins -Competition between HahMLV and Gag-PolMLV potentially reduces Cas packaging per particle -MLV protease may non-specifically cleave SpCas9 and reduce activity -Cas9 and sgRNA co-produced in the same cell	-Any type of CRISPR/Cas system can be packaged -Demonstrated for SpCas9 and dCas9-VPR
VEsiCas [207]	-Efficient Cas9 and sgRNA packaging -Very simple and easy-to-use fusion of Cas9-VSV-G and sgRNA-expressing constructs -Efficient, on-target genome editing -Tested in vivo	-Use HEK293T, a transformed cell line -Generated EVs are not exosomes; their properties and interaction with target cells need to be determined -Tissue-specific targeting upon systemic delivery has not been investigated -Quantity needed and quality of VEsiCas remain to be investigated -Composition of VEsiCas and co-packaging of potentially toxic proteins is not clear -Cas9 and sgRNA co-produced in the same cell	-Can be potentially expanded to clinically relevant EV-producing cell lines -Any type of CRISPR/Cas system can be packaged
Gesicles [208]	-Transfer Cas9:sgRNA RNPs -Efficient genome editing in target cells -Simple packaging system	-Use HEK293FT, a transformed cell line -Evidently less effective than NanoMEDIC -Cas9 protein half-life is reduced -<1% of produced gesicles contain RNPs -Carry-over of producer proteins and RNAs is possible -Use potentially toxic A/C heterodimerizer -Cytotoxicity and immunogenicity have not been studied -Not tested in vivo -Cas9 and sgRNA co-produced in the same cell -No tissue-specific targeting reported	-Potentially consist of a vesicle population mixed with cell waste as evidenced by increased gesicle formation following transfection

**Table 3 ijms-21-07362-t003:** Mechanisms for packaging RNAs into EVs.

Type of Packaging	Mechanism	Type of RNA	Advantages	Disadvantages	Used Previously for sgRNA Targeting?
Insertion of exosome-targeting motifs	-miR451 stem loop and its structural mimics [216]	-miRNA -May be suitable for sgRNAs	-Many thousand-fold enrichment in different cell types	-Enrichment is cell type-specific -Inefficient	No
	-EXOmotifs: GGAG in the 3′-half of RNA [217] -C/UCCU/G anywhere in RNA [217] -CTGCC motif [218] -Depend on hnRNPA2B1	miRNA	-Exosome-specific motifs	-Never used to load sgRNAs -Requires several motifs -Enrichment in EVs may depend on trans-acting factors, sequence context, secondary and tertiary structures -Efficacy is unclear	No
	-Insertion of HIV sequences -A2RE sequences present in Gag and vpr ORFs [219] -Depend on hnRNPA2B1	-Short RNAs	-Exosome-specific motifs	-Has never used for programmed loading -Efficacy unclear -May be cell-type specific	No
	-Secretion motifs: -ACCAGCCU -CAGUGAGC -UAAUCCCA	-RNAs -Non-coding RNAs	-Exosome-specific motifs	-Motifs may not be sufficient for transporting RNA into exosomes -May be cell-type specific -Requires a combination of different motifs	No
	-AnxA2-interacting motifs [220] -Putative binding motif is 5′-AA(C/G)(A/U)G	mRNAs	-Exosome-specific motifs	-Requires high-order RNA structures for interaction -May require two AnxA2-binding motifs -Depending on AnxA2 protein, may be cell-type specific	No
GEDEX or stochastic packaging [204]	-Stochastic packaging	sgRNAs	-Efficient -Proven delivery in vitro and in vivo in several disease models -Suitable for mass-scale production	-Packaging is most likely cell type-specific -Produced in transformed cell lines -Safety issues	Yes^209,226^
Insertion of exosome-targeting motifs [202]	-Ψ+-RGR HH ribozyme-sgRNA-HDV ribozyme-pA -Ψ+ interacts with expressed HIV Gag protein to package HH-sgRNA-HDV into exosomes -HH and HDV self-cleave to release sgRNA -HIV Tat/Tar interaction is required for EV packaging	RNAs	~4-times more efficient at loading sgRNAs than stochastic loading from U6-sgRNA -Proven as a NanoMEDIC CRISPR-loading platform	-Requires several HIV proteins -HIV Gag and Tat proteins are associated with oxidative stress and may be toxic to cells [Oxidative Stress during HIV Infection: Mechanisms and Consequences]	Yes, as a component of NanoMEDIC^208^
EXOtic RNA packaging devices [221]	-Archaeal ribosomal protein L7Ae binding to C/Dbox RNA structure -Fused CD63-L7A3 interacts with 3′-UTR C/Dbox-containing RNA -Number of C/Dbox-moieties affects efficacy -Connexin 43 (Cc43) acts as a cytosolic RNA delivery helper	-mRNAs -May be suitable for short RNAs	-Efficient -Can be adapted for sgRNAs packaging -Shown to be functional in human MSCs	-Release of RNA in target cells needs to be clarified	No
TAMEL platform [222]	-EV-enriched protein fused with an RNA-binding domain (MS2 protein dimer) -EV-enriched proteins: Lamp2b, CD63, Hspa8	-mRNA -May be suitable for short RNAs	-Very efficient for RNA loading -Can be adapted for sgRNAs packaging	-Efficiency of RNA release unclear (no mRNA translation seen in target cell) -RNA is not released or is degraded by lysosomes	No
LID RNA binding [223]	-Palmytoylation sequence-EGFP-CIBN CYR2-mCherry-MCP BFP-miR-21Sponge-6×MS2-PolyA	-miRNA -May be used for short RNAs	-Very efficient (~14-fold enrichment) -Reversible -Can be adapted for sgRNAs packaging	-Requires blue light illumination (may be toxic to the producer cells)	No
Chemical RNA modification [224,225]	-Covalent conjugation of RNAs to hydrophobic moieties: -Docosanoic acid (efficient packaging) -Cholesterol (efficient packaging) -Tocopheryl succinate (vitamin E) (most efficient packaging) -TEG linker for attaching hydrophobic moiety to RNA is most efficient -RNA must be modified to resist phosphatase and nuclease	-Shown for siRNA -Can be used for sgRNAs and other short RNAs	-Thousands of copies packaged per vesicle -Very efficient -Very useful for mass-scale production -No need to be intracellularly produced, can be mixed with EVs	-Substantial portion is attached to the surface of EVs -Unknown which RNAs (surface-bound or luminal) are functionally active -Highest loading EV efficacy shown is 43% -The level of loading may be cell type-specific -Release of RNAs from EV membrane is not clear	No
RNA transfer by making hybrid exosomes [193,226,227]	-Incubation of exosomes with RNA-loaded liposomes for 12 h at 37 °C	-DNA -May be useful for RNA loading	-Incubation makes most exosomes form hybrids with liposomes -Very efficient -No need for intracellular RNA production -Very useful for large-scale production	-Long-term incubation at 37 °C may deteriorate RNA -Hybrid liposomes are toxic to cells (modification of liposomes is essential)	-Yes -Tested for CRISPR/Cas9-expressing DNA plasmids
Physical methods	-Electroporation of EV-producing cells [98]	Any nucleic acids	-Very efficient and reproducible -Can deliver both Cas and sgRNAs into EVs simultaneously	-Damaging to EVs (large holes in membranes)	-Yes -Not suitable for clinical applications
	-Sonicating EVs for RNA loading [228]	-siRNA -Suitable for other short RNAs	-No significant aggregation of RNAs or EVs -Very efficient	-Degradation of RNA with prolonged sonication -Damaging to EVs – impaired functional properties -May damage Cas proteins pre-packaged into EVs	No
	Heat shocking EVs [229]	-miRNAs -Suitable for other short RNAs	-Efficient RNA loading -Suitable for large-scale production	-RNA deterioration -May result in degradation of pre-packaged Cas proteins -Impairment of EV membranes	No
	pH gradient modification of EVs [230]	-miRNAs -Suitable for other short RNAs	-Efficient -Suitable for large-scale production	-Evident protein degradation in EVs (decreased total protein content)	No
	Freezing-thawing [193]	-Mostly used for engineering EV surfaces	-Likely results in CRISPR/Cas RNPs packaging	-Freezing-thawing may destroy CRISPR/Cas components -Damaging to EVs	No
	Extrusion [231]	-Disrupts EV membranes	-Potentially able to load sgRNAs into EVs -Resulting EVs are not toxic	-Damages EV membranes	No
Incubation with membrane permeabilizers	Saponin [228]	-Damages EV membranes	-Potentially able to load sgRNAs	-Saponin is a cytotoxic agent [232] -Saponin induces hemolysis [232]	No

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
