# Peer review of "Gene Editing by Extracellular Vesicles"

_ijms, 2020, doi:10.3390/ijms21197362_

Round 1

Reviewer 1 Report

The manuscript “gene editing by extracellular vesicle” aims to review the current state of the art on CRISPR/Cas9 delivery with a special focus on extracellular vesicles (EVs).

The authors are reviewing other methods of CRISPR/Cas9 delivery the EVs very superficial and/or too generalized. Statements that Cas9 could not be cloned into viral vectors or that nanotechnological methods would not suitable for CRISPR/Cas9-RNP delivery are incorrect and ignore published work in these areas. In addition, biological processes are described incorrectly. I advise the authors to give correct, more precise and less general statements to suit a scientific article.

Also the EV-based delivery of CRISPR/Cas9 is not described pointedly. The authors compile a large set of methods, many have not been performed with Cas9 and/or sgRNA yet. The packaging of RNPs – as indicated as the aim of this review- was not addressed at whole, only packaging of Cas9 and sgRNAs are reviewed separately.

Altogether, I do not recommend this manuscript for publication. Please see my detailed comments below.

All abbreverations need to be introduced

Introduction of nucleic acids to mammalian is called transfection, transformation is performed in bacterial cells. Therefore throughout the whole manuscript transformed cells need to be changed to transfected cells.

Line 31: sgRNA is the abbreviation for single guide RNA, not single-guided RNA

Line 40: please specify “other safety issues” and give a reference

Line 42ff:

The authors state that one platform for the delivery of all CRISPR/Cas9 approaches is needed. I do not think that this can be stated in such a generalized way. It would not be a large problem if a specific and efficient platform for each single approach would be developed.

Line 62ff:

The authors should describe which delivery methods for DNA-based systems exist. They do not mention a single method here. In addition, they state that ”the large molecular size […], exceeds the packaging capacity of viral vectors and thus hampers their use.” This is not correct. There are several adenoviral and lentiviral vectors encoding for different Cas9-approaches. This needs to be corrected and also explained in detail.

Line 72f:

For this statement the authors need to add references. In the current form it is a mere postulate.

Line 87:

RNP-approaches do not need production of CRISPR/Cas9, please rephrase.

Line 92:

This has not been stated above.

Line 93ff:

This is not correct. There are reports (e.g. Wei et al., 2020, Systemic nanoparticle delivery of CRISPR-Cas9 ribonucleoproteins for effective tissue specific genome editing, nature communications and several more) on nanotechnological methods to deliver Cas9 as RNPs. In lines 104f the authors even cite one study.

Line 130:

What is meant with “along with others reviewed in this manuscript”

Line 217ff:

I have several issues with the sentence “Upon allogeneic transplantation of EVs, the issue of MHC histocompatibility may induce graft-versus-host disease” and the following passage.

1) The cited reference does not include any information on extracellular vesicles, allogenic transplantation or graft-versus-host disease.

2) Graft-vs.-host disease would imply that the EVs attack host cells, which seems rather unlikely to me. If this is the case, it needs further explanation and citations. The next sentence rather describes host-versus-graft reaction, which would also make more sense. Nevertheless, this would rather lead to a clearance of EVs from the body than to a disease. In addition, there are studies that use EVs to reduce graft-vs-host disease as the authors even state a few sentences later themselves. The authors need to perform a more founded reviewing of the literature and basic immunological principles here and maybe rewrite this section (lines 212 – 235) to make correct and straight-forward statements.

3) Reference 97, cited in the following sentences, does not refer to transplated EVs but states that transplanted mesenchymal stem/stromal cells generate EVs. First, this describes a completely different approach and must not be mixed up with the injection of EVs. Second, the cited study does not give any indications that the host’s immune system may damage or destroy cells which engulf EVs. If there is a work showing this, the authors need to cite it here; otherwise, this is only a weak speculation.

Lines 236 - 256

The following section does not deal with the efficacy in contrast to what is implied by the first sentence.

Lines 244ff:

The sentence “Cells typically have limited proliferating capacities” again is too general. Immortalized cells and the later mentioned stem cells have a high proliferating capacity. The authors should also define “transformed cells” more precisely. The reduction of stems cells to MSC, as implied by the sentence in line 254 also does not include the whole spectrum of stem cells.

Lines 263f:

This sentence implies that all EVs originating from MSCs promote angiogenesis and carry potential cancerogenic cargo. I assume – and the next sentence supports this -  that this assumption again is too general. I would assume that this depends on the subsets of EVs that are produced. THis should be reviewed more precisely.

Lines 274ff:

The authors mention problems that of course need to be solved but will an essential part of GMP production. This is more a principle issue than an EV specific safety issue.

Lines 294ff:

Here it would be interesting, if also other organs than the mentioned (liver, spleen, lung, gastrointestinal tract) are meant by this statement. A citation is missing here. Only in the next paragraph (line 305) a review is cited.

Lines 363 ff:

Enrichment of lipids, click-chemistry and PEG-coating are no genetic engineering methods, as indicated by the section heading.

Line 484:

The sentence seems to be lacking a word (motif?).

483 till end:

Remove all methods that have not been performed with Cas9 or sgRNAs. The topic of the review is gene editing with extracellular vesicles but the major part of methods described have not been applied in the context of gene editing. This also applies for the tables. In addition, no methods that have not been mentioned in the text should be added. The tables are far too full and contain much off-topic information. In my eyes, the tables could be removed completely.

Lines 500 ff:

The authors should mention the packaging of sgRNAs

Line 508:

Which distribution is meant? And the reason for the assumptions in the sentence does not become clear.

Lines 512ff:

The method should be explained in more detail, the expression of Gag-Pol-Pro, cell-line used, efficacy, … The authors also need explain why they assume a cleavage of cellular proteins, if the carry over is low and whether there is evidence for a non-specific cleavage of Cas9 by MLV-protease. They also should note that the sgRNA is also packaged by this method.

Figure 1:

Remove A, this was not performed with Cas9, remove C for the same reason and because there is not part in the text matching this figure. Fibure 1B is missing the cleavage of Gag and Cas9 and the Gag-Pol must be Gag-Pol-Pro. VSG-G needs to be explained and in the legend the statement the Cas9 is fused to Gag-Pol is wrong.

Line 535:

Change VesiCas to VesiCas9

Line 543:

If the efficacy is mentioned for one method, it needs to be mentioned for the others too.

Line 615:

Which protein is meant? Cas9?

4.2.1.2 and 4.2.1.4

Explain the methods in detail. This are the only method listed, which have been applied to sgRNAs, so better focus on this one.

Lines 677-679:

Are there studies supporting these assumptions?

Lines 680-708:

There is no relation to the topic of the review. Remove.

In addition, the induction of innate immune response by sgRNAs must not be mixed up with the mortality in transfected cells. Innate immunity refers to the whole organisms and not to the cellular mechanisms after transfection.

4.2.2.1

Has this been performed with sgRNAs?

Author Response

Please, see our detailed response to the Reviewer's critique in the attached file.

Reviewer 2 Report

This is an excellent and up to date review presented by Kostyushey et al. summarizing the current knowledge on the use extracellular vesicles for the delivery of CRISPR/Cas9. Several approaches were discussed.  The authors discuss also, the potential  limitations and the results of the clinical trials.

Minor points:

The main focus of the review is the delivery of CRISPR systems, rather than other gene editing tools (ZFNs, Talen….)  so may be the title should be modified accordantly.

Line 62 …..”with poorly controllable intracellular synthesis of CRISPR/Cas components with ensuing increase in off-target activity” . The reference 38 Fu et al., do not associated the  off target effect with the delivery of the CRISPR CAS9 as DNA coding sequence. The same observation related to reference 39.

 …..Rather to the specificity, the main limitation of the delivery of CAS9 as DNA sequences, is the high toxicity associated with the delivery of high amount of DNA into the cells. Maybe alternative references should be taking in account.

Line 211 safety of EVs

One of the major issues related to the use of allogeneic EVs, is related to the major histocompatibility complex (MHC) molecules present in the EVs.  The author state that transplantation of allogenic EVs may induce graft-versus host disease. I believe that rather to induce GVHD the implementation they will suffer a rejection, by the host immune system of the infused EVs ¿is that correct?  May be the authors can be more explicit on that point.

Line 221 “ … This is particularly true for therapies involving mesenchymal stem/stromal cells (MSCs), which originally were thought to express no or minimal amount of MHC antigens on their surface. A compelling line of evidence has indicated that injecting MSCs into allogeneic donors induces immune response, resulting in rapid clearance of MSCs and low efficacy of therapeutic interventions”

This paragraph seem to contain contradictory ideas…..one of the advantage to work with MSCs, is to reduce immune response, but according to the text is seem to be the opposite.

Line 225, “In contrast, several studies demonstrated that EVs are not immunogenic or much less immunogenic than EV-producing cells per” however the author mention these two articles:

Wahlund, C. J. E. et al. Exosomes from antigen-pulsed dendritic cells induce stronger antigen-specific immune responses than microvesicles in vivo. Sci. Rep. 7, 17095 (2017).

Hiltbrunner, S. et al. Exosomal cancer immunotherapy is independent of MHC molecules on exosomes. Oncotarget 7, 38707–38717 (2016).

In the first articles Wahlund et al., perform a comparison between exosomes and microvesicles (MV), concluding that the exosomes are able to induce an immune response. Maybe the authors should be clearer in that point.

Line 493: “Previous studies demonstrated that fusing POI with ubiquitin motifs is sufficient for their 493 enrichment in EVs”, the author refers to a general review, it will be more informative to refer to the original paper, were the incorporation of ubiquitin motif enhance the enrichment of the EVs.  Line 609: “Overexpression of RNA or protein inside a producer cell does not guarantee…” the reference 204 (Chen et al do not support the idea that the overexpression do not guarantee the packaging into EVs…), may be the authors should substituted with the alternative one.

Author Response

Please, see our detailed response to the Reviewer's comments in the attached file.

Round 2

Reviewer 1 Report

The authors have responded to my concerns and erased points. I think one problem isthat the authors evoke expectations or impressions by some statements they seemingly do not want to evoke. I have explained this in more detail in the response to the specific responses. Clarification of these points is in my eyes essential.

I have three major concerns remaining.

(1) The authors state in their response that they aim to point out methods that could be used for CRSIPR/Cas9 delivery. From the abstract (lines 15f, revised version) and the introduction (e.g. lines 133ff, revised version) I expected that the focus would lie on successfully established methods. Reading the passages after their explanation, I can sense their meaning, however, it was not conceivable at first sight. Moreover, I would still not expect from these passages that the main part of section 4 should dedicated to such potential methods. Therefore, I first recommend to clarify explicitly, that next to established methods, also methods with a high potential to be used in the future are discussed. Second, it is necessary to highlight the existing methods. Therefore, I recommend to change the structure of section 4. Start with already established methods and go on with potential methods. This will make the state of the art visible. 

(2) I still do not approve the extensive tables. I esteem the work the authors have put into their inquiries on these methods, but the large amount of different methods that are not described in the text cannot stand on their own. As it is now, it remains a list of methods that have been applied on protein or RNA delivery by EVs. I agree that some of these methods may be used in future for CRISPR/Cas9 approaches. But if the authors aim to focus on future aspects they need more than just tables. I suggest to stick to the methods described in the text.

(3) Even though the authors state that they did not aim to focus on RNP delivery, this is an essential point for CRISPR/Cas9-based genome editing. It will most likely not reach sufficient efficacy to deliver Cas9 and sgRNA in separate EVs. Therefore, the focus of the review schould lie on the delivery of RNPs by EVs.  

Detailed responses:

I agree to the responses I do not mention here further.

Response 1: The authors now phrase this more carefully now, so the sentence is no longer incorrect. However, I still think it would be more impartial to state that saCas9+sgRNAs can be delivered by AAV-approaches. Other viral approaches could be mentioned here too, along with the safety issues.

Response 2: I appreciate the addition of the sentence and also the sentence mentioned in response 10. However, my remark mainly referred to the sentence “High positive charge and molecular mass (>160 kDa for S. pyogenes Cas9) make CRISPR/Cas RNPs unsuitable for methods of nanotechnological packaging and protein delivery.” (lines 93ff in the first version). In the revised version (lines 105ff) the term “traditional” was added, however, I still think this is not a correct statement. I would assume that liposomes, gold nanoparticles, etc. – which have been applied to CRISPR/Cas9 delivery - belong to the traditional methods of nanotechnological packaging. If this was not meant here, the authors need to define “traditional” more explicitly.

Response 3: I appreciate that the authors now mention in the text, which methods have not been applied yet in CRISPR/Cas9 delivery. I also value that the authors aim to describe methods which could be used in future. I suggest to change the order of the described methods and first describe methods that have been applied already and then the others.

Response 4:

The authors claim that they „never stated the packaging of RNPs as the aim of this review”. However, in the abstract they state, “In this review, recent advances in developing vehicles for delivery of CRISPR/Cas in the form of ribonucleoprotein complexes are outlined.” (lines 15f). In the manuscript the state they would “In this manuscript, we […] review available technologies for engineering EVs and the recent progress in CRISPR/Cas RNPs delivery using synthetic nanoparticles and EVs, synthetic or naturally produced.” (lines 134f).

If they do not want to make this point in their review, they need to remove these sentences. From my point of view, this sentence evokes the impression that the review would deal with development of techniques for RNP-delivery and not for delivery of protein and sgRNA alone. Importantly, as mentioned above, I think it is necessary to put a focus on RNP delivery, if talking about CRISPR/Cas9-genome editing.

Response 6: I now see my misunderstanding here. 

Response 9: I completely understood that it was not the aim of this review to give an overview over all possible platforms. I never expected this. My remark pointed out my doubt that one platform for all would be needed as implied by: “(4) the lack of a universal CRISPR/Cas delivery platform that can be 50 utilized for a wide array of CRISPR/Cas systems Such a platform must allow use of CRISPR/Cas systems that are highly variable in size and molecular features; systems isolated from various species (e.g., Neisseria meningitides18, Streptococcus thermophilus, Streptococcus pyogenes, others like the recently described small CasX from Deltaproteobacteria); and engineered CRISPR/Cas21, like CRISPRa/i tools, CRISPR base editors and the PrimeEditing system.” (lines 50ff, revised version).

I suggest to remove this part.

Response 10: I am well aware that there are plenty of studies that show RNPs are most effective in CRISPR/Cas9-delivery. Nevertheless, a review should at least mention other methods that are used, this does not have to be in detail. In addition, the studies cited here are maybe not the best suited ones as they deal with bacteria and plants. There are enough studies in mammalian cells and organisms.

Response 16: Please refer to Response 2.

Response 23: I agree that this is an important issue, however, it will also apply for other systems. Therefore, I suggested omitting this passage.

Response 27: I appreciate that the authors aim to give an outlook on future methods. As suggested in Response 3, please reorganize this passage to highlight methods that already have been applied to Cas9 and afterwards describe methods with future potential. This will give a higher impact to the already existing methods.

As pointed out before, the tables need to be reduced to the methods described in the text.

Response 31: For Figures 1A and B as suggested in Response 3, please restructure the enumberation of methods to highlight existing Cas9-delivery methods.

Remove Figure 1C. If the authors think that this method is that important that it should be highlighted by a figure, they need to describe it in the text. Tables and figures are ment to support statements in the text and shall not stand on their own without reference to the text.

Response 35: As stated in Response 3 modify the description order to highlight existing methods. I do not agree with the authors that they gave an “exhaustive description”. The methods should be comprehensible for the reader of this review without referring to the cited reference.

E.g. 4.2.1.4: The method is not conceivable from their description and may be misunderstood in its shortness. Gag is not expressed endogenously in the cells but co transfected, the cells lines used for EV-generation is not mentioned, the removal of packaging signals is not directly linked to the ribozymes, etc. This method has been successfully applied to deliver CRISPR/Cas9 (both!) and should be valued with more than 6 lines. A minor point is that the authors might also mention the name of the method “NanoMEDIC” in the text, not only in the figure and the table.

Reference 36: I appreciate the alteration. It directly addresses my concerns that the passage was written more like a statement than like a discussion of the potential.